# Pressure as a Limiting Factor for Life

**DOI:** 10.3390/life6030034

**Published:** 2016-08-17

**Authors:** Rachael Hazael, Filip Meersman, Fumihisa Ono, Paul F. McMillan

**Affiliations:** 1Christopher Ingold Laboratories, Department of Chemistry, University College London, 20 Gordon Street, London WC1H 0AJ, UK; r.hazael@ucl.ac.uk; 2Earth Sciences Department, University College London, Gower Street, London WC1E 6BT, UK; 3Biomolecular & Analytical Mass Spectrometry Group, Department of Chemistry, University of Antwerp, Groenenborgerlaan 171, B-2020 Antwerpen, Belgium; f.meersman@ucl.ac.uk; 4Department of Applied Science, Okayama University of Science, 1-1 Ridaicho, Kitaku, Okayama 700-0005, Japan; fumihisaono@yahoo.co.jp

**Keywords:** extreme conditions, high pressure treatment, food technology, protein stability, cell wall biochemistry, bacterial survival, interplanetary transport, seeds, spores, *Artemia cysts*

## Abstract

Facts concerning the stability and functioning of key biomolecular components suggest that cellular life should no longer be viable above a few thousand atmospheres (200–300 MPa). However, organisms are seen to survive in the laboratory to much higher pressures, extending into the GPa or even tens of GPa ranges. This is causing main questions to be posed concerning the survival mechanisms of simple to complex organisms. Understanding the ultimate pressure survival of organisms is critical for food sterilization and agricultural products conservation technologies. On Earth the deep biosphere is limited in its extent by geothermal gradients but if life forms exist in cooler habitats elsewhere then survival to greater depths must be considered. The extent of pressure resistance and survival appears to vary greatly with the timescale of the exposure. For example, shock experiments on nanosecond timescales reveal greatly enhanced survival rates extending to higher pressure. Some organisms could survive bolide impacts thus allowing successful transport between planetary bodies. We summarize some of the main questions raised by recent results and their implications for the survival of life under extreme compression conditions and its possible extent in the laboratory and throughout the universe.

## 1. Introduction

Life is currently only known to have evolved on Earth, although it has been hypothesized that it could have emerged and might be present elsewhere in the universe [1]. In fact, our own expeditions into space might be in the process of transporting live or dormant organisms outside our planetary envelope, thus perhaps “seeding” life elsewhere. Until quite recently, life on Earth was considered to occupy a narrow range of warm, damp habitats near the planetary surface [2,3,4]. However, both active and dormant organisms are now known to proliferate in an ever-widening range of habitats in which they are exposed to extreme physical and chemical environments [5]. These include high or low temperatures, adverse chemical conditions, intense radiation and particle beam fluxes, and exposure to extremely high mechanical stress and strain rate conditions. The survival of organisms under extreme static or dynamic compression has implications for determining the possible extent of deep biospheres on Earth and on other planetary bodies, as well as in the laboratory and inside industrial pressure vessels used for preservation of foodstuffs, pharmaceuticals and agricultural products. Pressure inactivation in the food industry is primarily enabled in open/flow systems operating at several hundred MPa whereas the methods considered here involve closed systems operated at much higher pressures extending into the GPa range. These are where we examine pressure effects as ultimately limiting the survival of organisms. New findings reveal the greatly increased survival rate of simple and complex life forms during shock compression on a nanosecond timescale, supporting the hypothesis of transport between planetary bodies via bolide impact.

Unicellular prokaryotic organisms including bacteria and archaea are the simplest and most adaptive organisms that survive and thrive in some of the harshest environments on Earth [6]. Extremophiles or “superbugs” colonize hot geysers and submarine oceanic vents, under Arctic conditions, in acidic, alkaline and salt lakes, survive radiation exposure and high vacuum conditions [7,8,9,10,11]. These and other more complex organisms have been found within the ultrahigh pressure environments of the deepest ocean trenches, and extending to several kilometers below the oceanic and continental crust [12]. For example, at these depths the pressure can attain several hundreds to thousands of atmospheres (1 kbar = 100 MPa ≈ 1000 atm). On Earth the hydrostatic pressure gradient is approximately 10 MPa/km^−1^ in the oceans while those in the rocky crust increase to approximately 15–28 MPa/km^−1^. The temperature rises rapidly with depth and this could provide the main parameter limiting the extent of the deep biosphere [4,13]. Although it is difficult to quantify the exact values of extreme pressures that might be encountered by biological organisms on other planetary bodies it is clear that pressures in the 100–1000 MPa range and above could be experienced at lower temperatures [14,15,16] (Figure 1).

## 2. Pressure Stability Limits for Biological Macromolecules and Their Complexes

It might be thought that a reductionist approach could be helpful to fix the absolute limits for life survival under extreme compression, by determining a set of pressures above which the constituent macromolecules can no longer exist in a functional state. However, laboratory results reveal that this principle does not operate as simply as might be expected. The pioneering experiments of Bridgman [17] first revealed protein denaturation at pressures in the 700–800 MPa range. Irreversible protein denaturation due to pressure application is now well documented, and it typically shows an onset above 200–500 MPa at room temperature, completed by ~800 MPa. Dissociation of protein complexes that are critically important for biological functioning occurs at lower pressure, by around 200–300 MPa. Reduced functionality of membrane proteins or detachment of these proteins from the membrane typically occurs as a function of increasing pressure. However, structural changes in the membrane can protect membrane proteins from total induced inactivation [18]. In addition, laboratory studies of the complex phospholipid structures that form the primary structures determining cell walls indicate that these undergo phase transformations that would render them rigid and unviable for biological functioning above the 200–300 MPa range. In deep-sea organisms an increase in unsaturated lipids incorporated into the cellular membrane occurs. High pressures also lead to the appearance of ordered ‘gel-like’ phases that can result in an increase in hydration of the lipid head group and preserve the lamellar membrane structure [18]. Inclusion of other molecules including polysaccharides into the membranes renders these resistant to low temperature treatment and leads to cryopreservation strategies that might also contribute to high pressure survival of the cell envelope structure [19,20]. These key results suggest that exposure of organisms to beyond a limiting pressure range extending up to a few hundred MPa, should define the extent of life survival conditions. However, that prediction is not borne out by laboratory results, or by the documented existence of prokaryotes, as well as even more complex organisms under deep Earth conditions, at pressures extending into the GPa range.

## 3. Actual Results for Bacterial Survival into the GPa Range

Food scientists have long known that although there are few survivors from an initial bacterial sample by 700 MPa, some survivors persist beyond this commonly used “Pascalization” threshold [21]. A first report that bacteria might survive in a viable state to GPa pressures was presented by Sharma et al. using in situ determinations of metabolic activity in a diamond anvil cell [22]. Survivability of certain bacteria has now been demonstrated to extend into the 2–3 GPa range by culturing survivors recovered after high pressure treatments [23,24]. Here it was demonstrated that survivors that had been previously exposed to high pressure conditions could survive to even higher pressures. Those results contain important information about bacterial survival mechanisms, as well as posing new questions and hypotheses concerning the actual upper pressure limits for cellular life forms.

First, it is obvious that the bacterial survival results indicate that the pressure stability of proteins or lipid cell bilayers, do not constitute a primary barrier to cellular integrity or the continued functioning of biological processes under extreme compression conditions, that well exceed the established stability limits of the biomolecules exposed under in vitro laboratory conditions. While there are several hypotheses for the modification strategies adopted by various organisms to survive extreme high pressure treatment, the fundamental biophysical or biochemical protection mechanisms are not yet understood [18]. They also lead to hypotheses that bacterial life might exist to extreme depth within “cold” planets with low geothermal gradients, with ultimate limits extending well into the GPa pressure range. The extent of such deep planetary viability of cellular organisms will extend to increasing greater pressures as they are cycled between depth horizons. However, such cycling processes, will likely be associated with energy transfers and a geothermal gradient, and it is not known how that might affect “bugs” exposed to the extraterrestrial high-P, T environments.

## 4. Water in Cells

One key issue concerns the high pressure behavior of water that is an essential biological component constituting approximately 70% of both prokaryotic and eukaryotic cells. Liquid water provides a dynamic medium in which molecules can diffuse and interact, and the structure and function of biomolecules are to a large extent determined by the properties of water [25]. It has long been assumed that water has to be in its liquid state to facilitate life. Pure H_2_O crystallizes at approximately 1 GPa to produce the dense ice-VI phase [26]. However, water inside cells contains dissolved ions and it interacts strongly with macromolecules within the nanoconfined environment [27]. Key questions to be answered concern the physical and chemical properties of intracellular water under extreme compression, to determine to what pressures the biological functions are maintained. Experimental studies of dynamic processes using techniques including optical spectroscopy, nuclear magnetic resonance (NMR) and quasi-elastic neutron scattering (QENS) are under way to examine live cells under high pressure conditions [22,28,29,30,31].

## 5. Timescales of Biological Processes

It is now established that viable prokaryotic cells can survive pressurization into the 2–3 GPa range under static compression conditions on a laboratory timescale, and that some fractions of the initial population can be recovered to form colonies at ambient pressure [22,23,24]. It is not yet known if the cells continue to function normally at high pressure, although in situ spectroscopic studies at lower pressure (to 200 MPa) indicate a dramatic slowdown in metabolic rates [20,30,31]. Such subsurface organisms have been termed “zombie cells”, where the slow rates of metabolism may mean that these cells may not reproduce on timescales that are comparable to the surface world [32]. We can hypothesize that functioning cellular life might be present to well beyond the current observed limit of 150 MPa on Earth [33], as well as to depths of many kilometers below the surface of “colder” planets. We can also hypothesize that one of the effects of extreme pressurization is to lead to a lower cell density and a slower evolutionary path.

It is also well known that cells of simple and complex organisms have established protective mechanisms against long periods of desiccation and adverse physical conditions by entering a cryptobiotic or dormant state [34,35]. These mechanisms could also play a part in maintaining essential biomolecules and their functional components in a viable form, to pressures that might extend well into the GPa range. That would permit survival of organisms exposed to extreme compression conditions over cosmological periods of time that might be resuscitated when conditions improve. Although the density of surrounding mineral phases increases with depth inside planetary bodies, there always remain intergranular spaces or inclusions within individual minerals that could accommodate such intact cellular organisms.

## 6. Complex Lifeforms

Initial beliefs held that the oceanic depths were devoid of life. However, expeditions have now fully demonstrated the presence of complex life existing within the oceanic abyss and also at depths to 2 km in mineshafts drilled into the continental crust [36,37,38,39]. High pressure effects on organisms ranging from *Artemia* cysts in a cryptobiotic state, to various spores and seed varieties, as well as the remarkable tardigrade creatures known as “water bears”, have been investigated using large volume techniques at 7.5 GPa, holding the samples at pressure for timescales extending from a few hours to several days [40,41,42] (Figure 2). Remarkably, most of the pressurized organisms resume their normal biological function once they are recovered to ambient conditions. It is obvious that the mechanically resistant coatings of seeds and other multicellular envelope structures protect the interior structures from compressive stress, but those effects remain to be fully quantified. It is known that seed coatings can resist high fracture stresses and rupture at low moisture content [43]. *Artemia* eggs hatch upon exposure to water and the nauplii show normal swimming behavior. Likewise, water bears exposed to pressurization become motile without apparent effects from the extreme compression [44]. Some seeds do show changes in their subsequent growing behavior, indicating that some pressurization effects on the biochemical apparatus have been perceived. The results lead to hypotheses that complex multicellular life forms might exist in a functionally active state at pressures into the GPa range, or in hibernation with a surrounding protective envelope around the cellular components, to pressures extending to tens of GPa, on Earth as well as elsewhere in the universe.

## 7. Compression on a Rapid Timescale

Laboratory experiments using traditional pressurization techniques have noted some dependence of the survival of organisms dependent on the timescale of the pressure exposure. Mostly those results can be explained in terms of biological processes such as the accumulation and expulsion of damaged cellular components [45]. However, dynamic shock compression techniques that allow probing of events on nanosecond timescales reveal further mechanisms for survival of organisms to even higher pressures. Increased survival rates are recorded for wild type and pressure-adapted bacterial strains at 2 GPa, and some bacteria have been reported to survive following exposure to shock pressures as high as 78 GPa, with a rise time on the order of ns, and a pulse duration of μs [46,47]. Here, it becomes important to consider the timescale of the mechanical relaxation response of the bacterial cell envelope. For compression timescales lower than approximately 1 ms the cell walls exhibit a solid-like “glassy” response and this may protect the internal biochemical structures from the full effects of the shock [48]. Here, an upper pressure limit would be fixed by the fracture strength of the bacterial cell wall, that may extend even into the megabar range (~100 GPa) for the short timescale events [45]. In contrast to the bacterial results, shock studies on seeds and spores into the 1–5 GPa range result in lack or delay of germination and altered behavioral or growth patterns [42,49,50,51]. The difference between unicellular and organisms with a complex cellular structure is not known. The results lead to the hypothesis that at least some simple organisms may survive interplanetary transport and delivery to a new host via impact events providing the shock temperatures remain sufficiently low [52,53,54]. Finally, recent work has established the production of amino acids via hypervelocity shock synthesis from precursor molecules contained within cometary ices [55]. Those results lead to possibilities that creation of at least some of the building blocks of life itself might have occurred during high pressure impact events.

## 8. Conclusions

Although the highest pressure to which organisms are known to be exposed on Earth does not exceed 150 MPa, laboratory results indicate that at least a fraction of the population of bacteria or complex life forms can survive exposure to pressures far above this threshold. The observed effects depend strongly on the experimental conditions. Nevertheless, the data suggest that pressure represents a limiting factor for life, albeit at values beyond those experienced on Earth. It raises a number of questions, especially as life forms are found to survive pressures at which individual cellular components are thought to be in a dysfunctional state. It also touches upon the very fundamental question of how to define life and opens perspectives on the probability of life elsewhere in the universe.

## Figures and Tables

**Figure 1 life-06-00034-f001:**
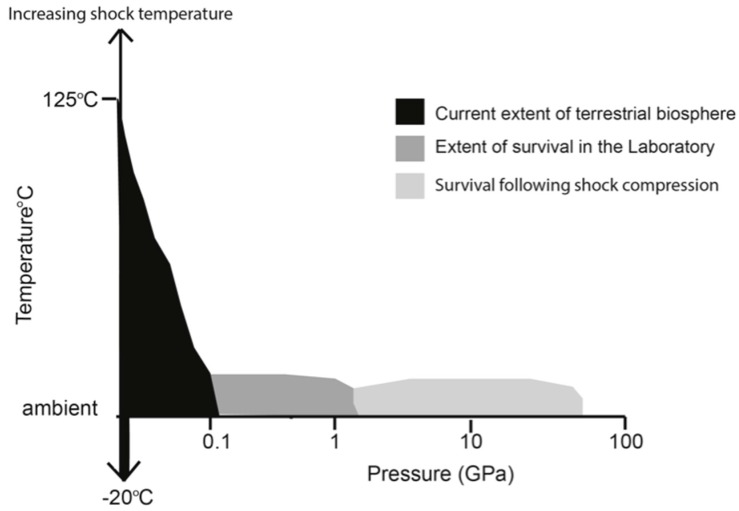
Cartoon illustrating the current extent of the survivability of organisms. This comprises both the habitable zone of the terrestrial biosphere to the important work of laboratory experiments, which extend current field sampling sites and expands the current physical limitations for life.

**Figure 2 life-06-00034-f002:**
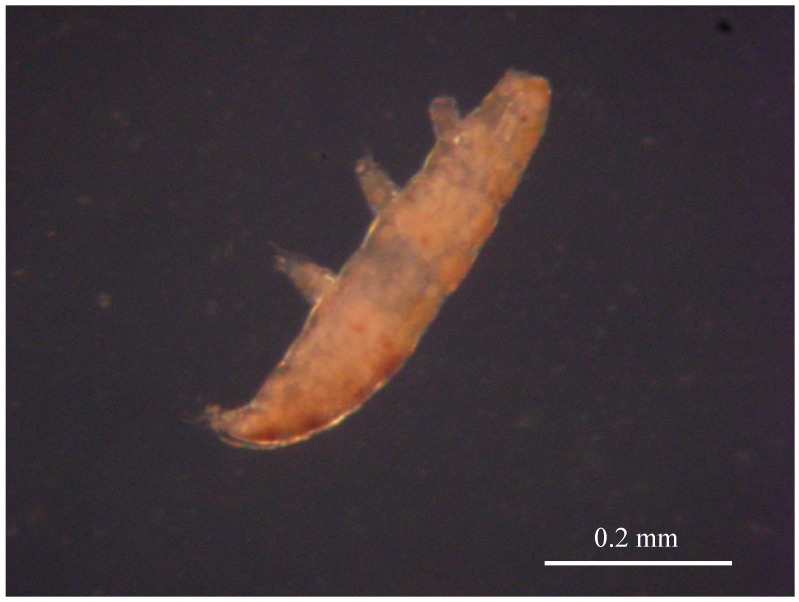
One of the five tardigrades which could tolerate exposure to 7.5 GPa for 12 h in a dormant (cryptobiotic) state, and then be revived by addition of water. The animal could move its legs, although slightly less energetically than other samples exposed for shorter times [40]. The resuscitated organisms were shown to survive for at least 7 days following the exposure to high pressure conditions [40]. A video documenting the remarkable recovery and swimming activity of this post-pressurization tardigrade, from which the still image is extracted, is shown in the Appendix A and was obtained by Prof. Fumihisa Ono (Physics) in collaboration with Dr. Masayuki Saigusa (Biology) at Okayama University, with help from undergraduate students in both their laboratories. The tardigrades for the study were mainly collected by Mr. Taro Uozumi.

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
