# Peer review of "Pressure as a Limiting Factor for Life"

_life, 2016, doi:10.3390/life6030034_

Round 1
Reviewer 1 Report
Please see MS pdf with handwritten notes.
Author Response
We would like to thank Reviewer one for their helpful comments. We have made the changes as requested. Please find attached the updated manuscript.

Reviewer 2 Report
I think that this concept paper is very interesting and will be useful to people from a broad range of research fields. I think that it should be published subject to a few minor points as described below.
Page 2, lines 50 – 60: It would be very useful in this paragraph and throughout to manuscript to differentiate clearly between pressures experienced in the ocean, pressures found within the earth’s upper crust and those thought to exist extraterrestrially.
Page 3, lines 73 – 77: It’s important to stress that while most proteins / lipids are rendered non-functional by the pressures described, there are certain modifications that can be made to these biomolecules to combat this. In general, I think it would be useful to mention some of the biomolecular structure strategies that organisms employ to survive under high pressure conditions (e.g. increased unsaturation / branching in lipid hydrocarbon chains).
Page 3, line 97: I think it would be useful to expand this statement a little. I’m not sure it’s correct to say that the stability of proteins and lipid membranes do not present a barrier to high pressure survival. These structures still need to maintain their fundamental integrity, and for continuing biological activity, they need to function - but organisms may employ modification strategies to allow this to happen in extreme conditions.
Page 5, line 155: This section would benefit from a little clarification. It’s not clear how seed coatings protect the interior structure. These coatings would be very deformable under the pressures mentioned and so the internal structure would be subjected to a pressure close to that applied to the seed. However, the coating may provide significant protection against ingress of water, other solvents and ions.
Page 6, line 181: It would be useful to note the timescale associated with the 78 GPa shock mentioned.
Author Response
We would like to thank reviewer two for their helpful comments. Please find below our responses and the updated manuscript attached.
Page 2, lines 50 – 60: It would be very useful in this paragraph and throughout to manuscript to differentiate clearly between pressures experienced in the ocean, pressures found within the earth’s upper crust and those thought to exist extraterrestrially.
We have clarified this issue at the beginning of the manuscript by defining the increase of pressure expected with depth on the Earth. Regarding extraterrestrial bodies we cannot give pressure values for each solar system body (or beyond), but we have further defined what are considered to be the relevant parameters to potentially habitable zones elsewhere.
Page 3, lines 73 – 77: It’s important to stress that while most proteins/lipids are rendered non-functional by the pressures described, there are certain modifications that can be made to these biomolecules to combat this. In general, I think it would be useful to mention some of the biomolecular structure strategies that organisms employ to survive under high pressure conditions (e.g. increased unsaturation / branching in lipid hydrocarbon chains).
We have added two sentences describing these effects in the text and have referenced our detailed recent and extensive review of these topics.
Page 3, line 97: I think it would be useful to expand this statement a little. I’m not sure it’s correct to say that the stability of proteins and lipid membranes do not present a barrier to high pressure survival. These structures still need to maintain their fundamental integrity, and for continuing biological activity, they need to function - but organisms may employ modification strategies to allow this to happen in extreme conditions.
We have slightly reworded our presentation of this topic and have expanded slightly on the statement. We have also referenced our review article where these effects are discussed in detail.
Page 5, line 155: This section would benefit from a little clarification. It’s not clear how seed coatings protect the interior structure. These coatings would be very deformable under the pressures mentioned and so the internal structure would be subjected to a pressure close to that applied to the seed. However, the coating may provide significant protection against ingress of water, other solvents and ions.
We note that no single or exact survival mechanism has been identified in the literature on this topic. We have added a statement in the text to specify that, at least for moisture-free seed coating, the mechanical rupture limit is high providing protection to the biological molecules in the interior. We have added a reference on fracture strength in seed coatings to illustrate this point.
Page 6, line 181: It would be useful to note the timescale associated with the 78 GPa shock mentioned.
This has been done.

Reviewer 3 Report
This is a very interesting article on a subject that is not easy to address. Obviously this article raises more questions than it gives answers. The reference to work on the inactivation of microorganisms by high hydrostatic pressures is interesting but I have not seen addressed the concept of closed system. I think we need to refer to these closed systems compared with open systems. In such closed systems the inactivation of microorganisms by the pressure obeys to Pascal's Law, this is not the case for open systems. The synergy between factors is also a topic that could be discussed in this article. These additions of topics would only strengthen this work
Author Response
We would like to thank reviewer three for their helpful comments. Please find our responses below and the updated manuscript attached.
This is a very interesting article on a subject that is not easy to address. Obviously this article raises more questions than it gives answers. The reference to work on the inactivation of microorganisms by high hydrostatic pressures is interesting but I have not seen addressed the concept of closed system. I think we need to refer to these closed systems compared with open systems. In such closed systems the inactivation of microorganisms by the pressure obeys to Pascal's Law, this is not the case for open systems. The synergy between factors is also a topic that could be discussed in this article. These additions of topics would only strengthen this work.
We note that while open systems are important for survivability of organisms in Pascalization environments, these do not lead to reflection on the ultimate high pressure limits of survival of biological systems that extend into the GPa range. That is the goal of this "hypothesis" paper. We have added a sentence in the text that mentions open systems, but they are not considered further in the context of our contribution.
